# The Nef Protein of the Macrophage Tropic HIV-1 Strain AD8 Counteracts Human BST-2/Tetherin

**DOI:** 10.3390/v12040459

**Published:** 2020-04-18

**Authors:** Sebastian Giese, Scott P. Lawrence, Michela Mazzon, Bernadien M. Nijmeijer, Mark Marsh

**Affiliations:** MRC Laboratory for Molecular Cell Biology, University College London, London WC1E 6BT, UK; s.giese@alumni.ucl.ac.uk (S.G.); scott.lawrence@ucl.ac.uk (S.P.L.); m.mazzon@ucl.ac.uk (M.M.); b.m.nijmeijer@amsterdamumc.nl (B.M.N.)

**Keywords:** BST-2, HIV-1, Nef, tetherin, macrophages, PBMC

## Abstract

Bone Marrow Stromal Cell Antigen 2 (BST-2)/tetherin inhibits the release of numerous enveloped viruses by physically tethering nascent particles to infected cells during the process of viral budding from the cell surface. Tetherin also restricts human immunodeficiency virus (HIV), and pandemic main (M) group HIV type 1s (HIV-1s) are thought to rely exclusively on their Vpu proteins to overcome tetherin-mediated restriction of virus release. However, at least one M group HIV-1 strain, the macrophage-tropic primary AD8 isolate, is unable to express Vpu due to a mutation in its translation initiation codon. Here, using primary monocyte-derived macrophages (MDMs), we show that AD8 Nef protein can compensate for the absence of Vpu and restore virus release to wild type levels. We demonstrate that HIV-1 AD8 Nef reduces endogenous cell surface tetherin levels, physically separating it from the site of viral budding, thus preventing HIV retention. Mechanistically, AD8 Nef enhances internalisation of the long isoform of human tetherin, leading to perinuclear accumulation of the restriction factor. Finally, we show that Nef proteins from other HIV strains also display varying degrees of tetherin antagonism. Overall, we show that M group HIV-1s can use an accessory protein other than Vpu to antagonise human tetherin.

## 1. Introduction

Human immunodeficiency virus type 1s (HIV-1s) are divided into four groups, M, N, O and P (Major, non-M/non-O, Outlier and ‘Pending the identification of further human cases’, respectively), each of which is believed to have originated from independent interspecies transmissions of simian immunodeficiency viruses (SIVs) into man. Group M, the first to be discovered, includes the principal global pandemic form of HIV-1. The success of group M viruses has been attributed to their ability to mount a potent anti-BST-2/tetherin (tetherin from hereon) defence in humans [1,2]. Tetherin is a 20 kDa glycoprotein that restricts a broad range of enveloped viruses by preventing their release from infected host cells [3]. Tetherin has two membrane-association domains: (1) a type-2 transmembrane domain at the N-terminus and (2) a glycophosphatidylinositol (GPI) linkage at the C-terminus. During viral budding, the tetherin N-terminal transmembrane domain remains embedded in the host plasma membrane (PM), but the C-terminal domain can be incorporated into the viral membrane, thus tethering nascent virions to the infected cell, preventing virus release and limiting viral spread [4]. Tethered virions can then be internalised and degraded. In addition to the full-length tetherin molecule (denoted long- or l-tetherin), a short isoform (s-tetherin) that lacks 12 N-terminal amino acid residues is generated by alternative translation initiation from a downstream start codon [5]. Both s- and l-tetherin physically retain nascent virions, but virus-associated l-tetherin can also trigger pro-inflammatory NFκB signalling that enhances viral restriction [5,6]. Human tetherin potently restricts HIV [7,8], and overcoming this restriction may have been a prerequisite for the pandemic spread of HIV-1 [2]. Indeed, a tetherin-mediated barrier to viral zoonosis has been suggested to be a product of the frequent interaction of viruses and antagonists driving their mutual evolution. This dynamic evolutionary arms race between virus and host is evidenced by the positive selection of transmembrane and cytoplasmic domain sequences in primate tetherins, some of which overlap with HIV-1 Vpu sensitivity [9,10].

Lentiviral antagonism of tetherin is highly conserved throughout primates. M group HIV-1s are thought to rely exclusively on their Vpu proteins to antagonise human tetherin and ensure efficient virus release from infected cells [11]. Interaction between the tetherin and Vpu transmembrane domains enables Vpu to displace tetherin from sites of viral budding and enhance its sorting to lysosomes where it is degraded [12,13,14,15,16]. Other HIV and SIV proteins also have some capacity to antagonise tetherin: For example, HIV-2 does not express Vpu but has adapted its envelope (Env) protein to enhance tetherin internalisation and overcome tetherin-mediated restriction [17] as has the Env protein from Tantalus monkeys [18]. Additionally, O group HIV-1s and many SIVs use their Nef proteins to antagonise the tetherin proteins of their respective hosts [19,20,21]. Further evidence for strong selection of tetherin antagonism has been seen in rhesus macaques infected with a Nef-deleted SIVmac239. The attenuated SIV developed mutations in the cytoplasmic domain of Env, enabling Env to counteract rhesus tetherin restriction, by sequestering tetherin away from sites of virus budding, and the virus to re-gain its pathogenic potential [22].

Nef is a 23 kDa HIV/SIV accessory protein that associates with the cytoplasmic leaflet of the PM via an N-terminal myristic acid moiety. The Nef protein of macaque-infecting SIV (SIVmac) recruits AP-2 complexes to macaque tetherin, which enhances clathrin-mediated endocytosis of tetherin and reduction of cell surface tetherin levels, though this does not lead to tetherin degradation [23]. SIV Nef proteins are unable to antagonise human tetherin using this mechanism [2,23,24] due to a five-residue deletion in the N-terminal cytoplasmic domain of human tetherin, suggesting these viruses adapted their Vpu protein for tetherin antagonism [1].

In light of these findings, the identification of AD8, an M group subtype B primary isolate that can replicate efficiently in monocyte-derived macrophages (MDMs) and peripheral blood mononuclear cells (PBMCs) in the absence of Vpu, is puzzling [25]. Here, we show that the AD8 Nef protein decreases the concentration of endogenous tetherin at the surface of primary human macrophages and reduces virus retention. Mechanistically, AD8 Nef promotes tetherin internalisation and perinuclear accumulation without causing tetherin degradation. Both N- and C-terminal domains of AD8 Nef contribute to reducing cell surface tetherin levels, which we observed for the long but not the short form of tetherin. Here, we show that some strains of M group HIV-1 encode a tetherin antagonist other than Vpu.

## 2. Materials and Methods 

### 2.1. Reagents and Antibodies

Tissue culture media and supplements were from Life Technologies (Paisley, UK), fetal calf serum (FCS) from PAA (Little Chalfont, UK), human AB serum from PAA and Sigma-Aldrich (Dorset, UK), and tissue culture plastic from Thermo Fisher Scientific (Waltham, MA, USA) and TPP (Trasadingen, Switzerland). DNA-modifying enzymes were from Promega (Southampton, UK) and chemicals from Sigma-Aldrich, unless specified otherwise.

Antibodies to HIV-1 p24/p55 Gag (38:96K and EF7), HIV-1 Env (b12 and 2G12) and HIV-1 Nef (2/81c/2j), as well as antiserum to HIV-1 p24/p55 Gag (ARP432), were obtained from the NIBSC Centre for AIDS Reagents (South Mimms, UK), and rabbit antiserum to BST-2/tetherin (cat. No. 11721) from the NIH AIDS Reagent Program (Germantown, MD, USA). Sheep antiserum to HIV-1 Nef was provided by M. Harris (University of Leeds, UK), rabbit antiserum to Vpu (U2-2) by K. Strebel (NIAID, Bethesda, MD, USA), and anti-vesicular stomatitis virus glycoprotein (VSVG) (P5D4) by T. Kreis (UNIGE, Geneva, Switzerland). Anti-adaptin γ (clone 88) was from BD Biosciences (Oxford, UK), polyclonal anti-BST-2/tetherin (B02P) from Abnova (Taipei, Taiwan), anti-CD81 (M38) from Abcam (Cambridge, UK), anti-HRP from Jackson ImmunoResearch (West Grove, PA, USA), anti-p65 from Santa Cruz Biotechnology (Heidelberg, Germany), anti-TGN46 (AHP500G) from AbD Serotec (Kidlington, UK), Alexa Fluor-conjugated antibodies from Life Technologies, highly cross-adsorbed DyLight 405-conjugated goat anti-mouse, as well as HRP-conjugated antibodies, from Thermo Fisher Scientific, and IRDye secondary reagents from Li-COR Biosciences (Cambridge, UK).

### 2.2. DNA Plasmids

The origins and descriptions of all DNA plasmids used in this study are listed in Appendix A.

### 2.3. Proviral Plasmids and Virus Stocks

The HIV-1 molecular clones AD8(+), AD8(−) and AD8(U_del2_) were provided by K. Strebel (NIAID, Bethesda, MD, USA) [26]. To obtain AD8(+)ΔNef, AD8(−)ΔNef and AD8(U_del2_)ΔNef, the third methionine codon in Nef was disrupted by introducing an adenine at nucleotide position 60, which causes a frameshift and a premature stop codon ten nucleotides downstream (see Appendix A for details).

HIV-1 AD8 stocks were prepared, as described previously [27]. Stocks of vesicular stomatitis virus glycoprotein (VSVG)-pseudotyped HIV-1 AD8 were prepared by co-transfecting HEK 293T cells with proviral and VSVG DNA for two days and then clearing the culture supernatants of cell debris by centrifugation.

### 2.4. p24 ELISA Assay

Gag p24 levels in cell-free supernatants from HIV-1-infected cells were quantified using the HIV-1 p24CA Antigen Capture Assay Kit (AIDS and Cancer Virus Program, National Cancer Institute, Frederick, MD, USA), according to the manufacturer’s instructions.

### 2.5. Single-Cycle Infectivity Assay

Single-cycle infectivities of HIV-1 AD8 stocks were determined, as described previously [27]. Single-cycle infectivities of VSVG-pseudotyped HIV-1 AD8 were determined by infecting HeLa cells with serial dilutions of virus stocks for two days and quantifying the proportion of infected cells by flow cytometry, as described below.

### 2.6. Cells, Transfections and Infections

HeLa, HEK 293, HEK 293T and HeLa TZM-bl cells were maintained in DMEM with GlutaMAX supplemented with 10% FCS, 100 U/mL penicillin and 0.1 mg/mL streptomycin. HEK 293 cells stably expressing s- or l-tetherin were provided by S. Neil (KCL, London, UK, [28]) and maintained in the presence of 100 μg/mL hygromycin B (Life Technologies).

Where indicated, HeLa and HEK cells were transfected for two days with TransIT-HeLaMONSTER (Cambridge BioScience, Cambridge, UK) or FuGENE HD (Promega), respectively, according to the manufacturers’ instructions. HIV-1 AD8 Env was co-transfected with HIV-1 Rev at a ratio of 3:1. For tetherin titrations, HEK 293T cells were seeded in 12-well plates and co-transfected with 400 ng proviral AD8 DNA and 400 ng tetherin-encoding DNA, or vector DNA, using FuGENE6 (Promega), according to the manufacturer’s instructions.

To infect HeLa cells with VSVG-pseudotyped HIV-1 AD8, cells were incubated with 0.3–0.4 IU/cell overnight, followed by a media change and a further 24 h incubation before harvest.

MDMs were prepared from peripheral blood mononuclear cells (PBMCs) isolated from buffy coats from healthy blood donors (National Blood Service, Essex, UK), as described previously [29], and differentiated in complete monocyte-derived macrophage (MDM) medium containing 10 ng/mL macrophage colony-stimulating factor (R&D Systems, Abingdon, UK) for two days. Seven-day-old MDMs were infected with HIV-1 AD8 (3 IU/cell) by spinoculation for 2 h at 1300 g and cultured for a further seven days. Unless specified otherwise, MDMs were used after 14 days in culture.

### 2.7. Western Blot Analysis

Cells were washed in PBS and lysed in non-reducing Laemmli sample buffer (4% SDS, 20% glycerol, 0.125 M Tris-HCl pH 6.8, bromophenol blue) for 10 min at 95 °C. Virus lysates were prepared by pelleting virus-containing culture supernatants through 20% sucrose cushions for 1.5–2 h at 100,000 g and 4 °C. Pellets were resuspended in Laemmli sample buffer and heated to 95 °C for 10 min.

Western blot analysis was performed essentially as described in [27]. Lysates were separated on SDS-polyacrylamide gels and transferred either to Immobilon-P or Immobilon-FL PVDF membranes (Millipore, Watford, UK) under semi-dry blotting conditions. Blots were quenched, incubated with primary antibody at 4 °C overnight, washed and incubated with either the appropriate HRP-conjugated secondary antibody or IRDye secondary antibody for 1 h at room temperature. After five washes, membranes were either incubated in SuperSignal West Pico/Dura/Femto Chemiluminescent Substrate (Thermo Fisher Scientific) and signals detected with Amersham Hyperfilm ECL (GE Healthcare Life Sciences, Little Chalfont, UK) or directly imaged on an Odyssey infrared imaging system (LiCOR).

All tetherin blots were performed under non-reducing conditions using the polyclonal BST-2 antibody B02P. For all other blots, 100 mM dithiothreitol was added to the lysates before gel electrophoresis.

### 2.8. Flow Cytometry Analysis

For flow cytometry analysis of cell surface tetherin or HIV-1 Env levels, cells were incubated for 1 h on ice in DMEM with GlutaMAX/FCS/penicillin/streptomycin or complete MDM medium containing rabbit antiserum to BST-2, 10 μg/mL polyclonal BST-2 antibody (B02P), or 10 μg/mL monoclonal anti-Env (b12) antibody. For HEK cells, 15 μg/mL anti-CD81 (M38) antibody was added during cell surface staining. Cells were washed once with ice-cold PBS, fixed in 4% PFA (TAAB, Reading, UK), scraped off the tissue culture dish, permeabilised with 0.1% saponin/1% FCS/2 mM EDTA/0.05% sodium azide/PBS, immunolabelled for 1 h with the additional primary antibody, washed three times in 0.1% saponin/1% FCS/2 mM EDTA/0.05% sodium azide/PBS, incubated for 30 min with appropriate Alexa Fluor-conjugated secondary antibodies, washed three times and analysed on an LSR II flow cytometer (BD Biosciences). For flow cytometry of MDMs, FCS was replaced by human AB serum, and 6 μg/mL human IgG was added during permeabilisation. For single-cycle infectivity assays of VSVG-pseudotyped HIV-1 AD8, infected HeLa cells were fixed, scraped off the tissue culture dish and processed for flow cytometry, as described above. Data were analysed using FlowJo software (TreeStar, Ashland, OR, USA). Transfected cells were identified by immunostaining for the transfected proteins or by GFP expression. Infected cells were identified by immunostaining for p24/p55 Gag. Relative cell surface tetherin/Env levels were calculated by dividing the median fluorescent intensities (MFI) of the transfected/infected cells by the MFI of the untransfected/uninfected cells of the same populations, unless specified otherwise.

### 2.9. Intracellular Tetherin Immunofluorescence

For intracellular tetherin immunofluorescence, transfected tetherin-expressing HEK cells were washed with PBS, fixed in 4% formaldehyde, quenched with 50 mM NH_4_Cl and permeabilised with 0.1% Triton X-100/0.5% BSA/PBS. Cells were labelled for 1.5 h with 10 μg/mL mouse anti-BST-2 (B02P), 1:100 rat anti-Nef serum, 2.5 μg/mL sheep anti-TGN46 and 1 μg/mL rabbit anti-p65 diluted in 0.5% BSA/PBS, washed in 0.5% BSA/PBS and incubated for 1 h with fluorescent secondary antibodies. Samples were washed and coverslips mounted in Mowiol. Confocal images were acquired with an inverted Leica TCS confocal microscope, 63× oil objective (NA 1.4) and LAS AF software, and processed using Fiji. An ImageJ macro was written for semi-automated analysis of perinuclear tetherin accumulation. Briefly, the macro segmented the area occupied by all cells (using p65 as a cytosolic marker), Nef-expressing cells (using Nef as a marker) and the area occupied by the trans-Golgi network (TGN) (using TGN46 as a marker). It then superimposed selections of the cell/transfected cell and TGN areas onto the respective tetherin stainings and measured the tetherin mean fluorescent intensities for both selections. To calculate TGN enrichment of tetherin, the TGN tetherin signal was divided by the tetherin signal from the residual cells and normalised to mock-transfected samples.

### 2.10. Cell Surface Tetherin Immunofluorescence

For cell surface tetherin immunofluorescence, MDMs were incubated for 1 h on ice in complete MDM medium containing 10 μg/mL anti-BST-2 (B02P), washed with ice-cold PBS, fixed in 4% PFA and quenched with 50 mM NH_4_Cl. For intracellular staining, cells were permeabilised with 0.1% Triton X-100/0.5% BSA/6 μg.ml^−1^ human IgG in PBS and labelled for 1.5 h with primary antibodies diluted in 0.5% BSA/6 μg.ml^−1^ human IgG/PBS, washed in 0.5% BSA/PBS and incubated for 1 h with appropriate combinations of fluorescent secondary antibodies. Samples were washed, DNA stained with 10 μg/mL Hoechst 33258 in PBS, and the coverslips mounted in Mowiol. Confocal images were acquired as above.

### 2.11. Statistical Analysis

Standard errors of the mean (SEM) and *p*-values are based on all replicates of at least three independent experiments. The *p*-values were calculated using the unpaired Student’s *t*-test, unless indicated otherwise.

## 3. Results

### 3.1. HIV-1 AD8 Nef Reduces Cell Surface Tetherin Levels

Evidence suggests that efficient Vpu-mediated counteraction of tetherin is required for HIV-1 spread [2]. Nevertheless, the ADA-derived M group HIV-1 molecular clone AD8 replicates in macrophages in the absence of Vpu [30,31]. Consistently, monocyte-derived macrophages (MDM) infected with AD8 containing the start codon mutation in Vpu [AD8(−)], or with an AD8 clone where Vpu was reconstituted [AD8(+)], release similar levels of virus measured as cell-free culture supernatant p24 (Figure 1A) or particle-associated p24 (Figure 3B and see [25]). These observations suggest that AD8 evolved a mechanism to antagonise human tetherin independent of Vpu. Indeed, in AD8-infected MDMs, we observed significantly lower levels of surface tetherin than in uninfected cells (Figure 1B,C).

To identify additional tetherin antagonists in AD8, we transfected candidate AD8 proteins, AD8 Env or Nef, or start codon repaired AD8 Vpu into HeLa cells, that express endogenous human tetherin. Two days later, the cell surface tetherin levels were quantified by flow cytometry. We found that start codon repaired AD8 Vpu reduced cell surface tetherin by around 85%, whereas AD8 Env had no effect (Figure 2A). Surprisingly, AD8 Nef also reduced cell surface tetherin by around 60% (Figure 2A).

To establish whether AD8 Nef also counteracts tetherin at expression levels reached during infection, we used previously described HIV-1 AD8 proviruses [25,30]. Since mutations in the Vpu start codon can enhance Env expression, we also used AD8(U_del2_), which carries an internal deletion in Vpu that abolishes Vpu expression [25,30]. For each of these Vpu mutants, we generated a Nef-negative variant AD8 ΔNef, which carries a premature stop codon in Nef (Appendix A). When we infected HeLa cells with vesicular stomatitis virus glycoprotein (VSVG)-pseudotyped HIV-1 AD8 for two days, the Vpu-negative Nef-expressing AD8(−) and AD8(U_del2_) still reduced cell surface tetherin by around 55% (Figure 2B), confirming that AD8 Nef antagonises tetherin at physiological concentrations. The Vpu-positive AD8(+) and AD8(+)ΔNef reduced cell surface tetherin by around 85%, whereas the Nef- and Vpu-negative AD8(−)ΔNef and AD8(U_del2_)ΔNef had no effect (Figure 2B).

Cell surface tetherin retains nascent virions at the surface of virus-producing cells. To test whether AD8 Nef-mediated tetherin antagonism reduces virus tethered to the cell surface, we infected HeLa cells with VSVG-pseudotyped HIV-1 AD8 and quantified cell surface Env by flow cytometry. As expected, the Vpu negative Nef positive AD8(−) and AD8(U_del2_) showed 1.7-fold less α-Env reactivity than the Vpu and Nef negative AD8(−)ΔNef and AD8(U_del2_)ΔNef, respectively (Figure 2C). Thus, AD8 Nef-mediated tetherin antagonism functionally reduces virus tethering to the cell surface.

### 3.2. HIV-1 AD8 Nef Reduces Cell Surface Tetherin Levels and Enhances Virus Release in Primary Macrophages

To confirm that AD8 Nef antagonises human tetherin in macrophages, we infected primary MDMs with the panel of viruses described above for seven days and quantified cell surface tetherin levels by flow cytometry. Similar to our observations in HeLa cells (Figure 2B), MDMs infected with the Nef-expressing AD8(−) and AD8(U_del2_) showed almost two-fold lower cell surface tetherin levels than cells infected with the Nef-negative AD8(−)ΔNef and AD8(U_del2_)ΔNef (Figure 3A). Thus, AD8 Nef also depletes cell surface tetherin in primary macrophages.

We next tested whether AD8 Nef-mediated counteraction of tetherin enhances virus release from MDMs. Considering the known effects of Nef on virus infectivity [26,32,33,34], we chose to monitor virus particle release by p24. While the highest levels of particle-associated p24 release were observed for cells infected with the Vpu-positive AD8(+) and AD8(+)ΔNef (Figure 3B right panels), AD8(−) and AD8(U_del2_) infected cells released more p24 than AD8(−)ΔNef and AD8(U_del2_)ΔNef-infected MDMs, respectively (Figure 3B right panels), indicating that Nef compensates for the loss of Vpu-mediated tetherin antagonism. Interestingly, western blot analyses of MDM lysates revealed that significantly less p24 was associated with AD8(−) and AD8(U_del2_) infected cells than with AD8(−)ΔNef and AD8(U_del2_)ΔNef infected cells (Figure 3B left panels). Since p24 is only found in mature HIV, and cell-associated p24 serves as a marker for retained virus, these observations show that more HIV is retained in the absence of AD8 Nef than in its presence. Overall, these findings demonstrate that on primary macrophages, AD8 Nef reduces cell surface tetherin and promotes virus release.

### 3.3. HIV-1 AD8 Nef Reduces Cell Surface l-tetherin Levels and Increases Virus Release

To examine whether AD8 Nef counteracts s- and l-tetherin, we transfected AD8 Nef into HEK cells stably expressing either s- or l-tetherin [28] and quantified cell surface tetherin by flow cytometry. AD8 Nef reduced the cell surface levels of l-tetherin by around 45%, but only caused a 10% drop in the cell surface levels of s-tetherin (Figure 4A). Thus, AD8 Nef primarily reduces l-tetherin expression at the cell surface.

We next investigated whether the AD8 Nef-mediated decrease in cell surface l-tetherin levels promotes virus release. We co-transfected proviral HIV-1 AD8 together with increasing amounts of s- or l-tetherin DNA into HEK cells for two days, then harvested the cells and cell supernatants and quantified virus release by ultracentrifugation of culture supernatants and western blot analysis (Figure 4B and Appendix A). At low concentrations of l-tetherin (20 ng transfected plasmid DNA), release of the Nef-positive AD8(−) and AD8(U_del2_) virus, assessed by p24, was significantly higher than that of the Nef-negative AD8(−)ΔNef and AD8(U_del2_)ΔNef (Figure 4B). The highest levels of p24 release were observed for the Vpu-positive AD8(+) and AD8(+)ΔNef (Figure 4B). Thus, HIV-1 AD8 Nef reduces the cell surface levels of l-tetherin and enhances virus release.

### 3.4. N- and C-Terminal Domains of HIV-1 AD8 Nef Contribute to the Reduction in Cell Surface Tetherin

The Nef proteins of many HIV-1 strains, including NL4.3, are thought to be inactive toward human tetherin. Indeed, NL4.3 Nef did not significantly decrease cell surface tetherin levels when we transfected HeLa cells and quantified tetherin by flow cytometry (Figure 5). To determine which AD8 Nef domains mediate tetherin antagonism, we generated chimeric Nef proteins that comprised of the N-terminal 85 residues of AD8 Nef and the C-terminal 122 residues of NL4.3 (AD8-NL4.3 Nef), or the N-terminal domain of NL4.3 and the C-terminal domain of AD8 Nef (NL4.3-AD8 Nef). When transfected into HeLa cells, both AD8-NL4.3 and NL4.3-AD8 Nef reduced cell surface tetherin levels by 45% and 30%, respectively, whereas AD8 Nef caused a 55% drop in cell surface tetherin (Figure 5). In addition, mutating the N-terminal myristoylation signal (G2A), previously shown to prevent Nef association with membranes [35], significantly impaired AD8 Nef’s ability to reduce cell surface tetherin levels (Appendix A). Thus, membrane-association is essential for, and both the N- and C-terminal domains contribute to, AD8 Nef-mediated antagonism of human tetherin.

### 3.5. HIV-1 AD8 Nef Enhances Tetherin Internalisation and Causes Perinuclear Tetherin Accumulation

When we infected MDMs with HIV-1 AD8 and analysed the cell lysates by western blotting, we did not find consistent differences between the total cell tetherin levels in AD8(−)/AD8(U_del2_) and AD8(−)ΔNef/(U_del2_)ΔNef infected cells (Figure 3B and Appendix A), suggesting that AD8 Nef does not promote tetherin degradation. To further investigate the mechanism by which AD8 Nef reduces cell surface tetherin, we transfected AD8 Nef into tetherin-expressing HEK cells. Confocal immunofluorescence imaging revealed that AD8 Nef-expression caused a perinuclear accumulation of l-tetherin that partially overlapped with the trans-Golgi network marker TGN46 (Figure 6A). No such intracellular accumulation was observed in AD8-Nef-transfected s-tetherin-expressing cells or mock-transfected cells (Figure 6A). ImageJ-based quantification confirmed that AD8 Nef expression caused significantly more perinuclear accumulation of l- than of s-tetherin (Figure 6A).

Intracellular tetherin accumulation may be caused either by impaired anterograde trafficking from the TGN to the cell surface or enhanced internalisation from the cell surface, or both. Since SIVmac Nef enhances macaque tetherin internalisation without causing its degradation [28], we hypothesised that HIV-1 AD8 Nef might employ a similar mechanism with human tetherin. To test this hypothesis, we transfected HeLa cells with AD8 Nef for two days, immunolabelled cell surface tetherin on ice, then warmed the cells to 37 °C to allow internalisation and quantified residual cell surface tetherin by flow cytometry. As expected, when tetherin was not internalised (0 min), cell surface tetherin levels were about 55% lower on AD8 Nef-positive cells than on Nef-negative cells in the same population (Figure 6B and Appendix A). However, when all graphs were normalised to this first time-point (0 min), we observed almost linear tetherin internalisation for the first 30 min after warm-up (Appendix A). Linear regression analysis revealed that the rate of tetherin internalisation was almost 1.5 times higher in AD8 Nef-positive cells than in Nef-negative cells (Figure 6B). Consistently, mutating one or both of the two AP-2 binding motifs in AD8 Nef slightly, but significantly, reduced Nef-mediated tetherin internalisation (Appendix A). Thus, AD8 Nef removes tetherin from the cell surface by enhancing its internalisation which, in part at least, contributes to perinuclear accumulation of the restriction factor.

### 3.6. Nef Proteins from Different HIV Strains Reduce Cell Surface Tetherin to Various Degrees

Finally, we examined whether the Nef proteins from other HIV strains reduced the cell surface human tetherin levels. We transfected IRES GFP constructs encoding Nef proteins from a range of HIV strains into HeLa cells and quantified cell surface tetherin levels two days later by flow cytometry. Consistent with our previous results, AD8 Nef reduced tetherin levels by around 55% (Figure 7). HIV O group Nef proteins (OMRCA and 13127K2), which have been described to antagonise human tetherin [19], reduced cell surface tetherin levels by around 70% (Figure 7). Interestingly, Nef proteins from HIV-1 M group strains—SF2, CH077 and CH198—as well as from the HIV-1 P group strain RBF168, reduced tetherin levels to a similar extent as AD8 Nef. By comparison, Nef proteins from the HIV-1 M group strains NL4.3, JR-CSF, the HIV-1 N group strain 2693BA and the HIV-2 strain 7312 showed smaller effects (Figure 7). An IRES GFP control that does not encode a functional Nef protein did not reduce cell surface tetherin levels (Figure 7). While this overexpression system can only provide an indicative measure of tetherin reduction across strains, the data suggest that Nef proteins from various HIV-1 strains possess some degree of anti-tetherin activity.

## 4. Discussion

Globally, M group HIV-1 infections account for more than 99% of HIV infections. The ability of these viruses to overcome the cellular restriction imposed by tetherin has been viewed as a key step in the global spread of HIV [2]; thus, understanding how M group HIV-1s counteract tetherin is important. Although it is known that HIV-1 Vpu is an efficient antagonist of human tetherin, we report here an M group subtype B HIV-1, the macrophage-tropic AD8, uses its’ Nef protein to antagonise this key restriction factor.

When the HIV-1 molecular clone AD8 was derived from the primary isolate ADA, it encoded an inactivating start codon mutation in Vpu [31]. As AD8 can sustain high levels of virus release from infected cells in the absence of Vpu, initial reports concluded that the viral Env protein possesses Vpu-like activity [25]. However, while a later study confirmed that high levels of Vpu-deficient AD8 are released, they failed to find evidence for Env involvement [36]. Consistent with this latter study, we show that AD8 Env does not reduce cell surface tetherin expression on HeLa cells (Figure 2A). Instead, we found that AD8 Nef decreases cell surface tetherin levels by over 50% (Figure 2A). In all our experiments, the anti-tetherin activity of start codon repaired AD8 Vpu was higher than that of Nef. However, we show that human tetherin downregulation from the cell surface by AD8 Nef is both specific and relevant for reducing virus retention and promoting virus release from primary MDMs, a natural target of AD8.

We also show that AD8 Nef downregulates cell surface levels of the long (l-) isoform of tetherin, in part at least, by increased endocytosis of tetherin, resulting in its intracellular accumulation (Figure 4, Figure 6 and Appendix A). The N-terminal cytoplasmic domain of l-tetherin contains serine-threonine-serine (STS) and conserved dual tyrosine motifs. The STS motif can be ubiquitinated [37], whereas the tyrosine motif can bind the clathrin AP-1 and AP-2 adaptor complexes [38]. Both motifs may cooperate with Vpu to induce tetherin internalisation and/or degradation [37,39]. S-tetherin lacks the twelve N-terminal residues that contain the STS and tyrosine motifs, which may explain why s-tetherin is insensitive to Vpu-mediated removal from the cell surface [5]. By analogy, the lack of AD8 Nef-mediated cell surface downregulation of s-tetherin suggests that AD8 Nef also cooperates with the STS and/or tyrosine motifs of l-tetherin to effect l-tetherin removal from the cell surface. Cooperation of AD8 Nef with a clathrin adaptor-binding motif in tetherin may explain why mutating AP-2 binding motifs in AD8 Nef alone has only a minor effect on its anti-tetherin activity (Appendix A). Alternatively, the twelve N-terminal residues of l-tetherin may be required for AD8 Nef binding. Interestingly, in contrast to SIVmac Nef proteins that downregulate the cell surface levels of both the l- and s-isoforms of macaque tetherin with equal efficiencies [28], the preference of AD8 Nef for l-tetherin mirrors that of HIV-1 O group Nefs [19].

Our data suggest that AD8 Nef potentiates virus release by lowering the cell surface levels of l-tetherin but does not significantly reduce cell surface or total levels of s-tetherin (Figure 4A and Appendix A). Nevertheless, AD8 Nef moderately increased virus release from provirus-transfected, s-tetherin-expressing HEK cells in the absence and presence of Vpu (Figure 4B and Appendix A), suggesting that AD8 Nef has some activity against s-tetherin. For efficient viral release, it is understood that SIVmac and HIV have alternative mechanisms to counteract s-tetherin [15,28]; thus, AD8 Nef conferring some activity towards s-tetherin, though unexplained, is reasonable.

Mechanistically, we have shown that a mutant AD8 Nef protein that does not associate with cellular membranes lacks anti-tetherin activity (Appendix A), and that lower surface tetherin levels are due to increased endocytosis and sequestration of tetherin in perinuclear endosomes (Figure 6). Nef proteins have been shown to modulate cell surface expression of a number of proteins, including MHC-I, CD4, CD8, CD28, and SERINC3/5 [32,33,40,41]. The best characterised of these Nef interactions is the one with CD4; HIV Nef proteins are known to bind to cell surface CD4, recruit AP-2 adaptor complexes and thereby promote CD4 internalisation by clathrin-mediated endocytosis. Similarly, we observed enhanced tetherin internalisation upon transfecting AD8 Nef (Figure 6B); it will be interesting to investigate whether the mechanism of tetherin antagonism by AD8 Nef resembles that of Nef-mediated CD4 downmodulation. Regardless, it is likely that AD8 Nef employs a different mechanism to Vpu to antagonise tetherin as, rather than at the cell surface, Vpu has been reported to interact with tetherin in TGN or endosomal compartments and promote lysosomal sorting of the restriction factor [3,13].

A previous study showed that HIV-1 O group viruses use their Nef proteins to counteract human tetherin [19]. Comparing that study to ours reveals a number of striking similarities: (i) Both O group and AD8 Nef only overcome low levels of tetherin, as found in primary cells; (ii) both Nef proteins primarily antagonise the long isoform of tetherin; (iii) O group and AD8 Nef enhance tetherin internalisation, contributing to its perinuclear accumulation. However, the amino acid residues key for O group Nefs’ anti-tetherin activity are not conserved in AD8 Nef. Instead, we show that both N- and C-terminal domains in AD8 Nef contribute to the reduction in the cell surface levels of human tetherin (Figure 5). Anti-tetherin activity has been reported for several M group subtype C HIV-1 Nef proteins [42]. In agreement with our findings, these Nef proteins downregulated cell surface l-tetherin, which correlated with enhanced virus release and replication. Although a mechanism for tetherin down-modulation was not identified, these authors reported that the Nef residues important for anti-tetherin activity were difficult to identify.

Considering that AD8 Nef is less efficient than Vpu in antagonising tetherin, it is unclear why AD8 would tolerate, or even select for, an easy-to-reverse start codon mutation in Vpu. Accumulating evidence suggests that macrophages are a physiologically relevant reservoir of persistent HIV and SIV infection [43,44]. It has been suggested that inactivation of AD8 Vpu may circumvent an as-yet-unknown function detrimental to HIV replication as the virus becomes compartmentalised in macrophages to establish a chronic infection [25]; AD8 Nef may then have adapted to antagonise human tetherin. Alternatively, the redundant anti-tetherin properties of Nef may have pre-existed the loss of Vpu expression, allowing this variation to be tolerated. Importantly, although the anti-tetherin activity of AD8 Nef is weaker than that of Vpu, it appears sufficient to overcome tetherin restriction of AD8 release from its primary target cells, i.e., macrophages, and it is interesting to consider whether this may have contributed to its tropism.

Regardless, whatever the advantage for HIV-1 to use Nef rather than Vpu as a tetherin antagonist, it appears that AD8 may not be the only virus exploiting this strategy. In this study, we found that the Nef proteins from a number of other HIV-1 strains, including two founder viruses, can reduce cell surface tetherin levels to varying degrees (Figure 7). Moreover, 1.24% of primary isolates in the HIV database encode a Vpu start codon mutation [45]; it will be intriguing to investigate whether some of these viruses use their Nef proteins to overcome tetherin restriction. Interestingly, Arias et al. observed that in viral strains with Nef anti-tetherin activity, the corresponding Vpu proteins displayed diminished or no anti-tetherin activity [42]. In conclusion, HIV-1 viruses appear to have retained some flexibility to use either Vpu and/or Nef to antagonize tetherin, thus ensuring that by one mechanism or another, the virus is able to antagonise this key restriction factor.

## Figures and Tables

**Figure 1 viruses-12-00459-f001:**
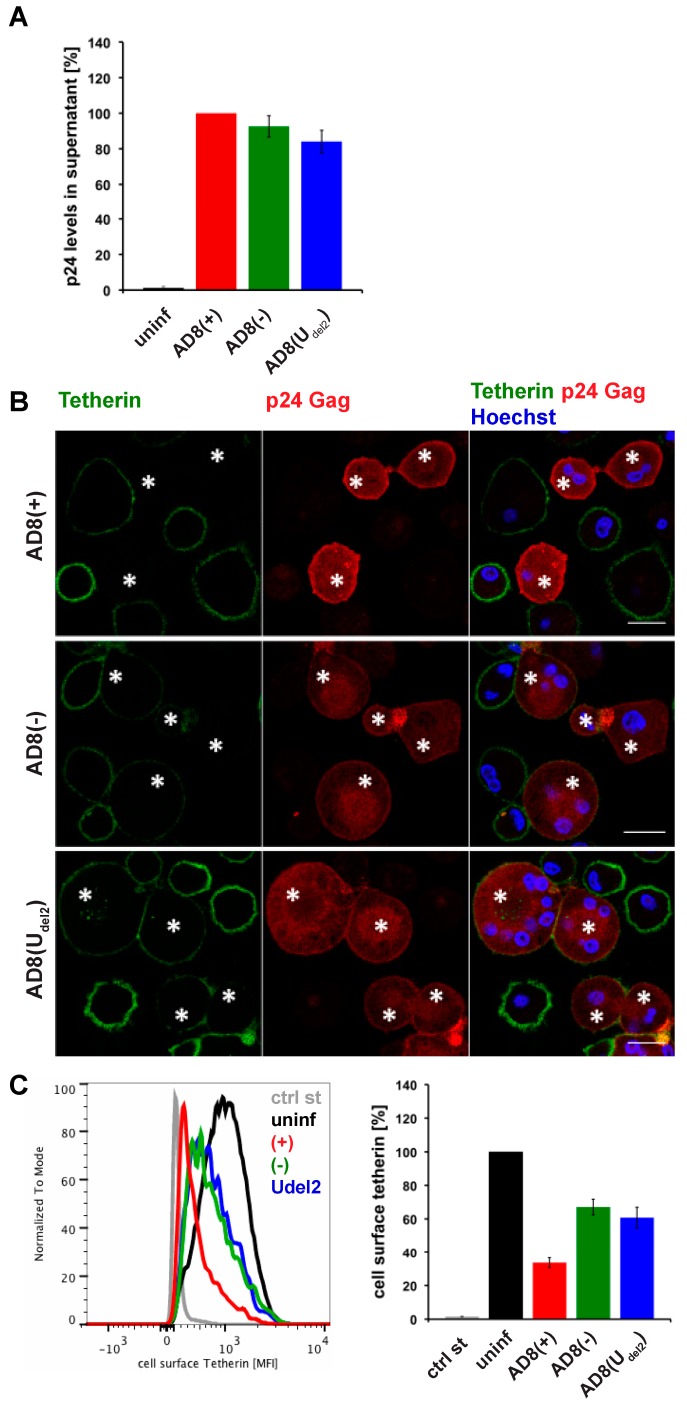
Human immunodeficiency virus type 1 (HIV-1) AD8-infected macrophages show low levels of cell surface tetherin and high levels of virus release in the absence of Vpu. (**A**) p24 Gag in cell-free culture supernatants from AD8-infected monocyte-derived macrophages (MDMs) were determined using ELISA; (**B**) AD8-infected MDMs were stained for cell surface tetherin and intracellular p55/p24 Gag. Asterisks mark infected cells. Single confocal sections are shown. Scale bars = 20 μm; (**C**) Cell surface tetherin levels of AD8-infected MDMs were analysed by flow cytometry. Uninfected MDMs were immunolabelled with anti-vesicular stomatitis virus glycoprotein (VSVG) as a staining control. The left-hand panel shows the result from a representative experiment; the right-hand panel shows the average relative tetherin mean fluorescence intensities of three donors ± standard error of the mean (SEM). Bars represent the relative means ± SEM of triplicate samples from three donors.

**Figure 2 viruses-12-00459-f002:**
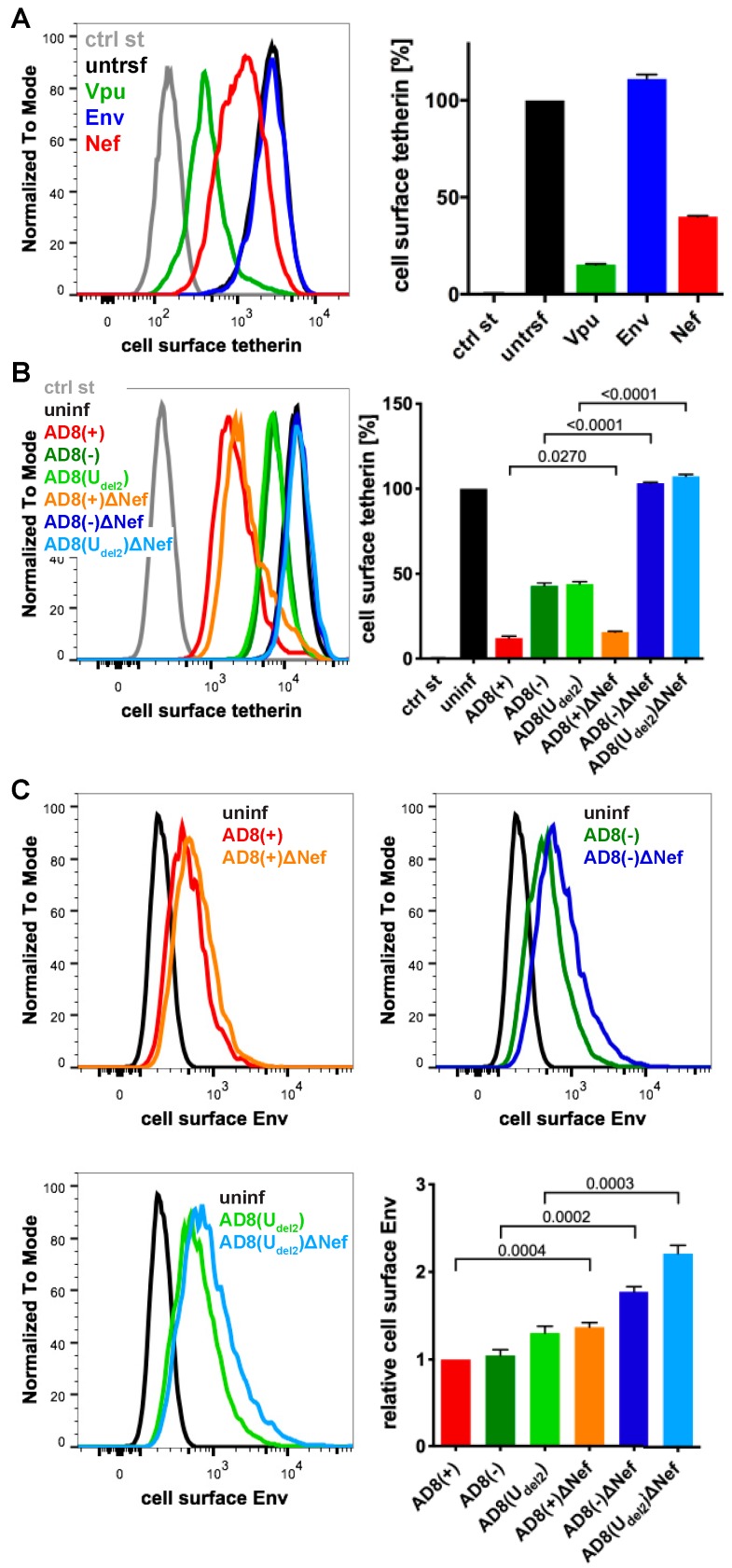
HIV-1 AD8 Nef reduces cell surface tetherin levels and tethered virions. (**A**) AD8 Vpu, Env, or Nef were transfected into HeLa cells for two days. Cell surface tetherin levels were quantified by immunolabelling the restriction factor on ice prior to formaldehyde fixation, cell permeabilisation, immunolabelling for Vpu, Env or Nef, labelling with fluorescent secondary antibodies and flow cytometry. Mock-transfected cells were immunolabelled with anti-VSVG as a control. The left-hand panel shows the result from a representative experiment; the right-hand panel shows the average relative tetherin levels from three independent experiments ± standard error of the mean (SEM); (**B**) cell surface tetherin levels of VSVG-pseudotyped AD8-infected or uninfected HeLa cells were quantified by flow cytometry. Uninfected cells were immunolabelled with anti-HRP as a staining control. The left-hand panel shows the result from a representative experiment; the right-hand panel shows the average relative tetherin levels from four independent experiments ± SEM with relevant *p*-values; (**C**) cell surface Env levels of VSVG-pseudotyped AD8-infected or uninfected HeLa cells were quantified by flow cytometry. The upper panels and lower left-hand panel show the results from a representative experiment; the lower right-hand panel shows the average relative Env levels from four independent experiments ± SEM with relevant *p*-values.

**Figure 3 viruses-12-00459-f003:**
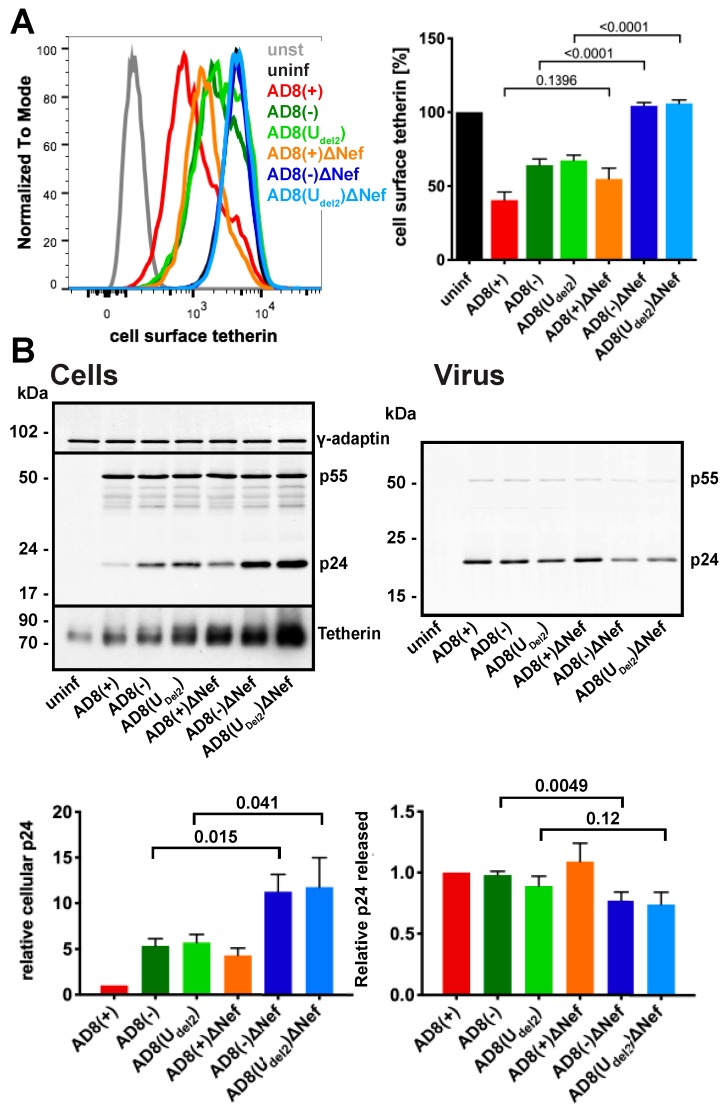
HIV-1 AD8 Nef antagonises tetherin in primary macrophages. (**A**) MDMs were infected with AD8 for seven days and cell surface tetherin levels quantified by flow cytometry. Uninfected cells were immunolabelled with anti-HRP as a staining control. The left-hand panel shows the result from a representative experiment; the right-hand panel shows the average relative tetherin levels from five donors ± SEM with relevant *p*-values; (**B**) MDMs were infected with HIV-1 AD8 for seven days. Virus and the corresponding cells were lysed and analysed by western blotting. The top panel shows representative western blots. The graphs in the bottom panels show the average relative cell (left) and virus (right) p24 levels from four donors ± SEM with relevant *p*-values.

**Figure 4 viruses-12-00459-f004:**
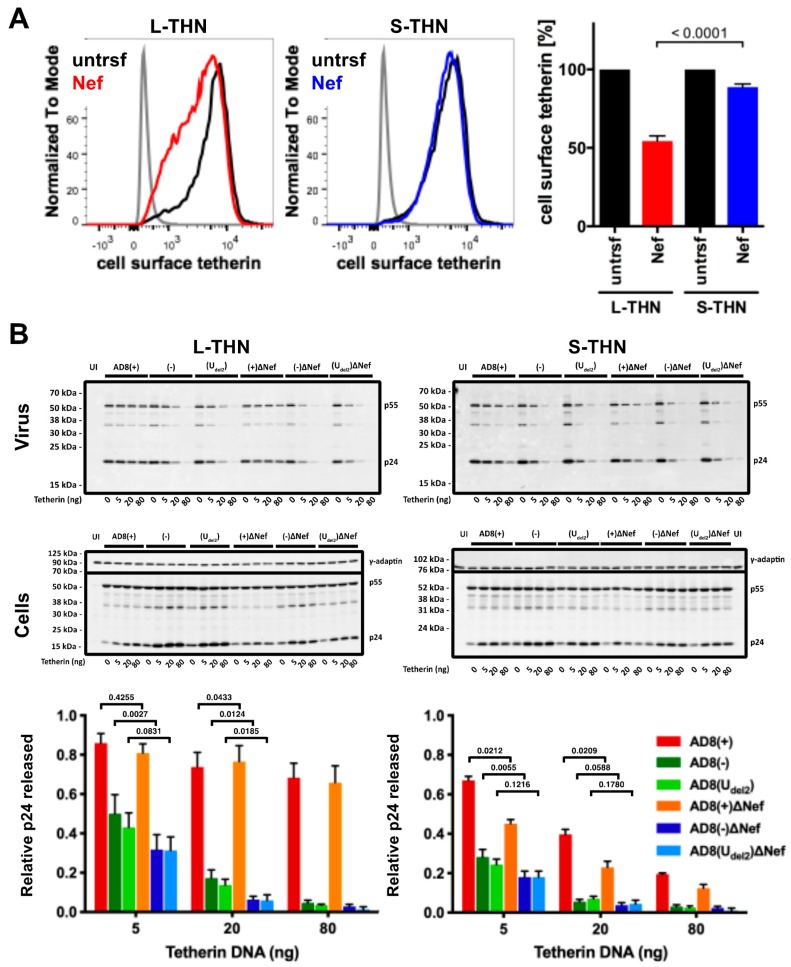
HIV-1 AD8 Nef antagonises the long (l-) isoform of tetherin and moderately increases virus release. (**A**) HEK 293 cells expressing s- or l-tetherin were transfected with AD8 Nef or a control plasmid for two days, and cell surface tetherin levels were quantified by flow cytometry. Mock-transfected cells were immunolabelled with anti-HRP as a staining control. To discriminate intact cells from debris, all cells were also stained and gated for cell surface CD81. The left-hand and central panels show the result from a representative experiment; the right-hand panel shows the average relative tetherin levels from three independent experiments ± SEM with relevant *p*-values; (**B**) HEK 293T cells were transfected with 400 ng of HIV-1 AD8 proviral DNA and increasing amounts of l- or s-tetherin-encoding plasmids. Virus and corresponding cells were harvested two days post-transfection, and all lysates were analysed by western blotting. Virus p24 Gag levels were normalised to cellular p55 Gag. The graphs show the average relative virus p24 Gag levels from at least four independent experiments ± SEM with relevant *p*-values. Tetherin expression was confirmed, as shown in Appendix A.

**Figure 5 viruses-12-00459-f005:**
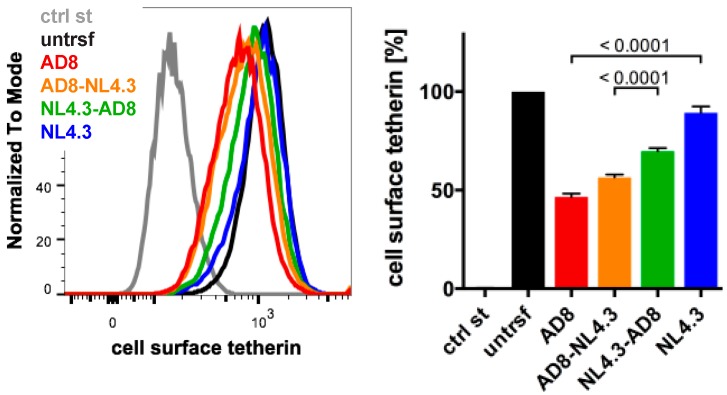
N- and C-terminal domains contribute to HIV-1 AD8 Nef’s anti-tetherin activity. HeLa cells were transfected with AD8 Nef, HIV-1 NL4.3 Nef or chimeric Nef proteins that contained the N-terminal 85 residues of AD8 Nef and the C-terminal 122 residues of NL4.3 Nef (AD8-NL4.3 Nef) or vice versa (NL4.3-AD8 Nef). Two days post-transfection, cell surface tetherin levels were quantified by flow cytometry. Mock-transfected cells were immunolabelled with anti-HRP as a staining control. The left-hand panel shows the results of a representative experiment; the right-hand panel shows the average relative tetherin levels from four independent experiments ± SEM with relevant *p*-values.

**Figure 6 viruses-12-00459-f006:**
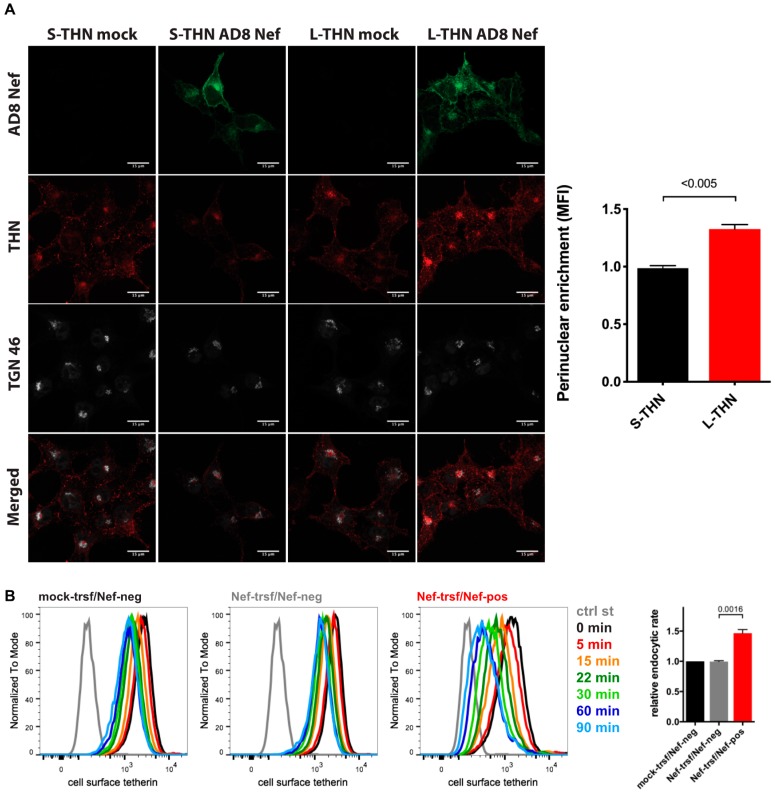
HIV-1 AD8 Nef antagonises tetherin by enhanced internalisation and perinuclear accumulation. (**A**) HEK cells expressing s- or l-tetherin were transfected with AD8 Nef or a control plasmid for two days then fixed and stained for intracellular tetherin. Single confocal sections from a representative experiment are shown. Scale bars = 15 μm. The graph shows accumulation of tetherin in a TGN46-positive perinuclear region in AD8 Nef-expressing cells. All values are normalised to mock-transfected cells, so that a value of 1 indicates no tetherin enrichment. The data show the average of three independent experiments ± SEM with relevant *p*-values; (**B**) AD8 Nef- or mock-transfected HeLa cells were immunolabelled for tetherin on ice and shifted to 37 °C for various times. Residual cell surface tetherin antibody was then revealed by immunostaining on ice with a fluorescent secondary antibody. Cells were fixed, permeabilised, immunostained for Nef and analysed by flow cytometry. The three left-hand panels show the result from a representative experiment for mock-transfected cells or AD8 Nef-transfected cells that did (Nef-pos) or did not (Nef-neg) express Nef. The right-hand panel shows the average endocytic rates from four independent experiments ± SEM during the first 30 min at 37 °C with relevant *p*-values. Endocytic rates were derived from the regression analysis shown in Appendix A.

**Figure 7 viruses-12-00459-f007:**
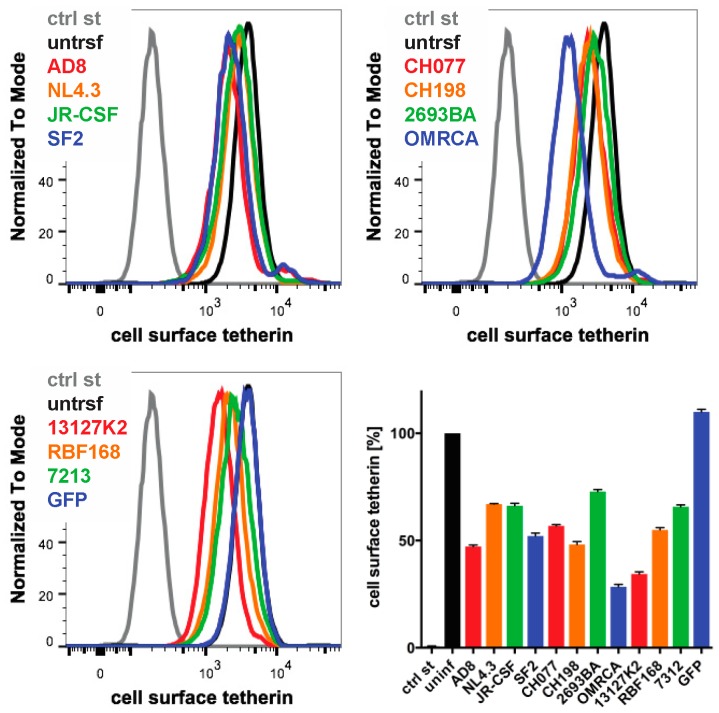
The Nef proteins of different HIV strains show varying degrees of anti-tetherin activity. HeLa cells were transfected with IRES GFP plasmids encoding Nef proteins from a number of HIV strains, and cell surface tetherin levels were quantified by flow cytometry. Mock-transfected cells were immunolabelled with anti-HRP as a staining control. The upper panels and lower left-hand panel show the result from a representative experiment; the lower right-hand panel shows the average relative tetherin levels from four independent experiments ± SEM.

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
