# Peer review of "The Nef Protein of the Macrophage Tropic HIV-1 Strain AD8 Counteracts Human BST-2/Tetherin"

_viruses, 2020, doi:10.3390/v12040459_

Round 1

Reviewer 1 Report

The manuscript by Giese and colleagues analyzes why AD8, despite being Vpu (-), can still antagonize human tetherin. A major concern with these experiments is that the authors rely on the release of p24 from cells as an indicator of virus release. No data was presented examining the levels of infectious virus released. Without this data, I feel the study is incomplete.

Major comments:

Comment 1: The AD8 strain of HIV-1 is known to replicate well in monocyte-derived macrophage cultures (MDM). While the authors have done a reasonable job showing that Nef may affect the release of Gag p24 from infected cells, p24 assays may or may not reflect the actual number of infectious units released. Since the effect of Nef on tetherin down-modulation is marginal, the levels of infectious virus need to be addressed or the findings of this study will be equivocal.

Comment 2: I have a major concern about whether Nef is all that important for virus release. For example, on the left panel of Figure 3B, the levels of p24 released (virus) for AD8(+) and AD8(+)ΔNef is nearly identical, which would argue against the author’s hypothesis. Additionally, on the same blot it appears that p24 levels released from AD8(+)- infected MDM, which have intact vpu and nef genes, are less than MDM infected with AD8(+) ΔNef. This would argue against a role of Nef in virus release. An alternative explanation could be that since the levels of tetherin in MDM are so low (Figure S1B), the virus may not require the anti-tetherin activities of Vpu or Nef to replicate efficiently.

Minor comments:

Line 16: “vpu” should be “Vpu.”

Line 21: “l-tetherin” should defined. The authors discuss l- and s-tetherin throughout the manuscript. The authors should define l-tetherin and s-tetherin and cite reference(s).

Line 31: “Bst-2" is a protein and is generally capitalized (BST-2).

Line 123-124: Under section 2.6, the authors state that HeLa cells were inoculated with 0.3–0.4 IU/cell. However, I could not find virus was titrated. This should be added.

Line 172: “NH4Cl” should be “NH4Cl.”

Lines 205-207: In Figure 1A, the authors discuss the levels of virus released from MDM inoculated with AD8(-) and AD8(+). They state that the two viruses “release similar levels of virus.” However, the authors show the levels of p24 released, not virus. Thus, the authors should change this statement to “release similar levels of p24."

Figure 2A: The authors should perform a statistical analysis for significance.

Figure 2B: In the right panel the authors state that the difference in the cell surface expression of tetherin in AD8(+) and AD8(+)ΔNef-infected MDM was statistically significant. I am having a hard time believing that this biologically significant since the authors do not measure infectious virus.

Figures 2A and B: It would help the reader (and reviewers) if the levels of cell surface tetherin expression are either discussed in the text or presented on the right panels.

Figure 2C, lines 240-243: The authors state that AD8(Udel2) showed 1.7 fold less ENV reactivity than AD8(Udel2)ΔNef yet the graph in Figure 2C indicates over two-fold. Please correct.

Figure 3A: In the right panel, the authors show that the difference in cell surface tetherin in AD8(+) and AD8(+)ΔN cells was not statistically significant, which argues against the hypothesis that Nef compensates for the lack of the Vpu. Please explain.

Figure 3B, bottom left western blot: Please explain why uninfected MDM have less tetherin than the various virus infections.

Figure 3B legend, line 283: Here the authors analyze the culture medium and cell lysates for viral protein. The sentence starting with “ Virus and .....” should be changed to “Virus containing culture medium and corresponding cell lysates were analyzed by Western blotting.” The rationale here is unless you have concentrated virus you are really analyzing the culture medium.

Figure 4: The authors state that they used HEK 293T cells expressing s- or l-tetherin. The description of these cells and/or their origin is not provided in the mmaterials and Methods section.

Figure 4B: The lower panels depict the “Relative virion p24." This should be changed to “Relative p24 released.” Virions are mature infectious viruses and what is measured are p24 levels.

Reviewer 2 Report

In this manuscript Giese et al. defined the role of HIV-1 Nef protein in antagonizing BST-2/tetherin function. This article is well writing, very detailed in its methods and presents new results in this area. This reviewer recommend this manuscript for publication with minor corrections.

Please delete space after references 1 and 2 in line 31.

Please add reference 20 (Schubert et al.) in line 205 after reference 19.

Please clarify that this sentence is referring to figure 3B "left panel" in line 269.

Please delete space in line 271.

Please clarify that this sentence is referring to figure 3B  "right panel" in line 273.

Please delete space in line 292.

Please check "G2A; figure S2A" in line 331.

Please delete space in line 414.

Please delete space in line 474.

Author Response

Response to Reviewer 2 Comments
We thank Reviewer 2 for his/her very positive views on our paper and suggestions for minor corrections, all of which have been addressed as indicated below.

Comments and Suggestions for Authors:
In this manuscript Giese et al. defined the role of HIV-1 Nef protein in antagonizing BST-2/tetherin function. This article is well writing, very detailed in its methods and presents new results in this area. This reviewer recommend this manuscript for publication with minor corrections. Please delete space after references 1 and 2 in line 31.

Response: This has been corrected.

Please add reference 20 (Schubert et al.) in line 205 after reference 19.

Response: This has been corrected.

Please clarify that this sentence is referring to figure 3B "left panel" in line 269.

Response: This has been corrected.

Please delete space in line 271.

Response: This has been corrected.

Please clarify that this sentence is referring to figure 3B "right panel" in line 273.

Response: This has been corrected.

Please delete space in line 292.

Response: This has been corrected.

Please check "G2A; figure S2A" in line 331. ? I suggest moving G2A earlier in sentence.

Response: This has been corrected.

Please delete space in line 414.

Response: This has been corrected.

Please delete space in line 474.

Response: This has been corrected.

Round 2

Reviewer 1 Report

The manuscript by Giese and colleagues analyzes why AD8, despite being Vpu (-), can still antagonize human tetherin. A major concern with these experiments is that the authors rely on the release of p24 from cells as an indicator of virus release. No data was presented examining the levels of infectious virus released. Without this data, I feel the study is incomplete.

Response: Measuring virion-associated p24 in the cell culture supernatant is an accepted method of looking at virus release. It measures the total virus (both infectious and non-infectious) released; As tetherin restricts the release of all virus whether infectious or non-infectious we consider this to be the most accurate reporter for tetherin restriction. We agree that virion-associated p24 does not measure infectious virus however, as Nef is known to impact on virion infectivity in addition to its effects on virus release (see references below), measurement of infectivity would be difficult to interpret in this study. For this reason, we have chosen to use an assay that directly measures total virus release. To avoid confusion, we have modified the text to indicate p24 release, rather than virus release.

Chowers, M.Y.; Spina, C.A.; Kwoh, T.J.; Fitch, N.J.; Richman, D.D.; Guatelli, J.C. Optimal infectivity in vitro of human immunodeficiency virus type 1 requires an intact nef gene. J Virol 1994, 68, 2906-2914.

Rosa, A.; Chande, A.; Ziglio, S.; De Sanctis, V.; Bertorelli, R.; Goh, S.L.; McCauley, S.M.; Nowosielska, A.; Antonarakis, S.E.; Luban, J., et al. HIV-1 Nef promotes infection by excluding SERINC5 from virion incorporation. Nature 2015, 526, 212-217, doi:10.1038/nature15399.

Zutz, A.; Scholz, C.; Schneider, S.; Pierini, V.; Munchhoff, M.; Sutter, K.; Wittmann, G.; Dittmer, U.; Draenert, R.; Bogner, J.R., et al. SERINC5 Is an Unconventional HIV Restriction Factor That Is Upregulated during Myeloid Cell Differentiation. J Innate Immun 2020, 10.1159/000504888, 1-11, doi:10.1159/000504888.

Reviewer’s Comment: While I agree with the authors that analysis of p24 will provide a measurement of p24 and likely virus particle release, it will not address the levels of infectious virus released. Additionally, one has to question the value of a restriction factor if it doesn’t reduce the amount of infectious virus. Further,it'slikely that the ratio of level of p24 released to the level of infectious virus released is likely to differ with each cell type. I don’t understand the statement that, “measurement of infectivity would be difficult to interpret in this study,” if they haven’t done the assays. Assay systems exist that can easily quantify the levels of infectious virus released. From the standpoint of the host, a restriction factor.

Regarding the Chowers reference cited, these investigators “quantified” p24 released. In the present manuscript, p24 doesn’t appear to be quantified.

Line 21: “l-tetherin” should defined. The authors discuss l- and s-tetherin throughout the manuscript. The authors should define l-tetherin and s-tetherin and cite reference(s).
Response: These abbreviations were defined, and references cited, when l- and s-tetherin where first discussed in the main body of text (section 3.3 starting from line 287). The term l-tetherin did appear in the abstract without clarification, so we have replaced ‘l-tetherin’ in the abstract with ‘long isoform of tetherin’.

Reviewer’s Comment: I believe it would be preferable to define the isoforms in the introduction during the discussion of tetherin/BST-2.

All other comments have been sufficiently addressed or clarified.  
